# The Comparison of Seven Models to Simulate the Transport and Deposition of Polydisperse Particles under Favorable Conditions in a Saturated Medium

**Zhike Zou [1], Lei Yu [1], Yalong Li [1,*], Shuyao Niu [2], Linlin Fan [1], Wenbing Luo [1] and Wei Li [1]**

[1] Agricultural Water Conservancy Department, Changjiang River Scientific Research Institute, Wuhan 430010, China; zz971278375@163.com (Z.Z.); ylei2833@163.com (L.Y.); fll901023@163.com (L.F.); luowenbing2005_0@126.com (W.L.); li-wei13@tsinghua.org.cn (W.L.)

[2] College of Hydrology and Water Resource, Hohai University, Nanjing 210098, China; n1148603335@163.com

[*] Correspondence: lyalong888@163.com; Tel.: +86-027-82828793

**Abstract:** Polydisperse particles are ubiquitous in both the natural and engineered environment, and the precise prediction of the transport and capture of polydisperse particles in a saturated medium is crucial. Several efforts (Yao model, RT model, TE model, MPFJ model, NG model, MHJ model, and MMS model) were developed to obtain accurate correlation equations for the particle capture probability (single-collector removal efficiency), but the applicability of the existing models to the entire porous medium and the retention characteristic of the polydisperse particles are still unclear. In this study, sand column experiments were undertaken to investigate the transport and capture processes of the polydisperse particles in the saturated medium. The mass density was employed to quantize the effects of particle polydispersity and incorporated into the depth-dependent deposition rate. The experimental results showed that the polydisperse particles formed a hyper-exponential retention profile even under favorable conditions (no repulsion). The excellent agreement between the results obtained from the MMS model and the experimentally observed results of the breakthrough curves (BTCs), as well as the retention profiles demonstrated the validation of the MMS model, as the correlation coefficient and the standard average relative error were 0.99 and 0.005, respectively. The hyper-exponential retention profile is caused by the uneven capture of the polydisperse particles by the porous medium. This study highlights the influences of particle polydispersity on particle transport and capture in a saturated porous medium.

**Keywords:** polydisperse particles; capture probability; deposition coefficient; model comparisons; uneven capture; hyper-exponential retention; favorable conditions

## 1. Introduction

Particle migration and capture in saturated porous media is prevalent in a wide range of environmental and industrial scenarios [1,2]. Included are concerns such as colloid-facilitated transport of viruses, bacteria, and nanoparticles in aquifers, suspended particle removal in sand filtration or wastewater treatment, and rock permeability reduction as a result of particle capture in drilling operations [3–6]. For all these above cases, it is critical to understand how particles are transported and captured in a porous medium and to predict particle removal through physically sound modeling [7–9].

Dispersed particles are ubiquitous in the groundwater environment and highly heterogeneously sized, ranging from the nanoscale up to fractions of a millimeter in natural and engineering conditions [10]. The transport processes of these dispersed particles through a porous medium and how they interact with the porous medium are governed by several different forces and transport mechanisms depending on the particle density and size [11]. These transport mechanisms mainly include Brownian motion, gravity, interception, attachment to media via attractive electrostatic forces, straining (pore throats are too small

for particles to pass), bridging, and trapping in dead-end pore throats [12]. The retention of particles larger than 10 μm is mainly accomplished by gravity, interception, and hydrodynamics, while Brownian motion is the most-prevalent transport mechanism for small colloids (<1 μm). For mean colloids (1–10 μm), all the acting forces and mechanisms can make a contribution to the transport and capture processes of colloidal particles [13]. These forces and mechanisms are complex and change greatly over time and space, which are inherently non-linear and operate jointly [14].

Over the past half century, the colloid filtration theory (CFT) has been modified at both the microscale and macroscale to model particle flow and capture in saturated porous media (collector) [1]. The transport and capture of colloids present a unique multiscale problem, and the theoretical underpinnings of the CFT consist of two fundamental components [15]: (1) a pore-scale analysis of particle flow and capture by a collector to define the removal efficiency of a collector and (2) upscaling from this pore-scale of particle motion analysis to a macroscale behavior. The first component is typically regarded as a progression of the transport and attachment stages, where the particle transport is governed by transport mechanisms and the attachment is approached due to the van der Waals attraction and electrostatic repulsion (DLVO theory) between a particle and the surface of a porous medium under favorable conditions [16]. Finally, a correlation equation for the frequency of particle contact with the medium surface, known as the single-collector efficiency, has been developed. The second component is conventionally achieved by the combination of a local mass conservation law with a kinetics equation for particle deposition [17]. This upscaling is primarily approached by continuum-based numerical models solving the advective and dispersion equation for solute transport in saturated porous media [18].

The first correlation equation was proposed in 1971 by Yao et al. [19], who assumed the correlation equation was the additivity of three analytical solutions for particle capture by an isolated sphere as a result of diffusion (D), gravity (G), and interception (I). However, this approach neglects the interplay between the three different capture mechanisms. Rajagopalan and Tien enhanced the Yao model in 1976 by conducting a numerical trajectory analysis of suspended particles under the actions of the van der Waals force (V) and the hydrodynamic retardation (H) in addition to the three mechanisms mentioned above [20]. Another commonly used correlation equation was developed by Tufenkji and Elimelech in 2004 [21], who performed Eulerian simulations in the Happel sphere and accounted for the coexistence of the three transport mechanisms, as well as the effects of the van der Waals force and the hydrodynamic retardation. In 2009, Ma et al. [22] introduced the hemispheres-in-cell model, which takes into account the effect of grain-to-grain contact points on the transport and deposition processes of a particle within a pore space. Nelson and Ginn [23] in 2011 developed a different correlation equation by conducting a large number of Lagrangian simulations in Happel sphere-in-cell porous media, and the new model simulated the concomitant presence of all the forces acting on the particles. Ma et al. [24] in 2013 extended the applicability of the correlation equation to low fluid velocity conditions by introducing a saturation factor on the basis of the particle mass transfer relationships. Messina et al. [25] in 2015 proposed a novel total flux normalized correlation equation, which included mixed terms that account for the mutual interaction of simultaneous transport mechanisms. These mixed terms could eliminate the negative effects of the use of additivity. The main features of the above seven correlation equations are summarized in Table 1.

Despite the significant efforts made by researchers in the development of correlation equations for the dynamic description of the particle behavior in saturated media, several mysteries remain because of the complex mutual interactions between particles and porous surfaces, especially when it comes to the effects of particle polydispersity on particle fate in a saturated medium [26,27]. Moreover, the similarity among the existing seven models naturally raises interest in their comparison and selection. However, few attempts have been made to explore the applicability of the existing correlation equations, especially for polydisperse particles, which are more prevalent in natural and engineered systems [28,29].

**Table 1.** The existing correlation equations for particle capture and their transport mechanisms.

| Acronym | Authors | Geometry | Transport Mechanisms |
| --- | --- | --- | --- |
| Yao model | Yao et al., 1971 [19] | Isolated sphere | Additivity of analytical solutions for D, G, I. |
| RT model | Rajagopalan and Tien, 1976 [20] | Happel sphere-in-cell | Additivity of analytical solutions for D, G, I, V, H. |
| TE model | Tufenkji and Elimelech, 2004 [21] | Happel sphere-in-cell | Numerical solutions for D, G, I, V, H. |
| MPFJ model | Ma et al., 2009 [22] | Hemispheres-in-cell | Numerical solution for D, G, I, V, H allowing for grain-to-grain contact points. |
| NG model | Nelson and Ginn, 2011 [23] | Happel sphere-in-cell | Numerical solutions for D, G, I, V, H for small particles at low velocities. |
| MHJ model | Ma et al., 2013 [24] | Modified hemisphere-in-cell | Numerical solutions for D, G, I, V, H at low fluid velocities. |
| MMS model | Messina et al., 2015 [25] | Isolated sphere | Numerical solution for D, G, I and the mutual interactions among the three. |

The overarching objective of this work was to model the transport and deposition processes of polydisperse particles in a saturated porous medium under favorable conditions. Laboratory sand columns filled with quartz sand of 0.503 mm were employed as experimental porous media. Polydisperse particles with a size range from 0.375 to 18.863 μm were made into a suspension and injected into the sand columns for breakthrough and final retention profile analysis. A comparison of the seven models (Yao model, RT model, TE model, MPFJ model, NG model, MHJ model, and MMS model) with the available experimental data was made. The specific objectives were to: (1) determine the effects of particle polydispersity on particle transport and deposition, (2) compare the performance of the seven existing models with the influent, effluent, and retention profile, and (3) simulate the transport and capture processes of polydisperse particles in the saturated porous medium.

## 2. Materials and Methods

### 2.1. Polydisperse Particles

The polydisperse particles were collected around the inlet of the Han River, and that could well represent the composition in reality. The material was ground, dried, and sorted repeatedly, ensuring a particle size range of 0–30 μm was obtained, since this specific size range is the most-commonly used. Particle size distribution (PSD) analysis was performed using dynamic light scattering (ZetasizerNanoZS90, Malvern Instruments Ltd., Malvern, UK). PSD analysis indicated the polydisperse particles ranged in sizes from approximately 0.375 to 18.863 μm, and the median size was around 2.932 μm (Figure 1). The particle polydispersity can be characterized by the particle mass density, namely the probability density of the particle size distribution [30]. The zeta potential of the particles was measured to be −24.3 mV in deionized water and values of −0.69 mV in 200 mM NaCl at pH 6.8 (25 °C). The specific density of the polydisperse particles was measured to be 2.53 g/cm$^3$.

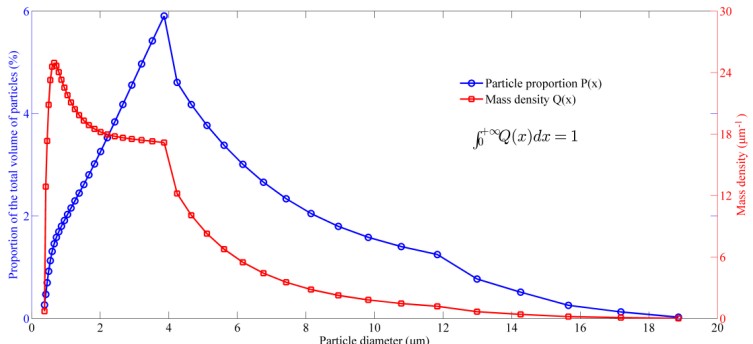

**Figure 1.** The proportion of the total volume of particles and the particle mass density. $Q(x)$ is defined as the particle proportion divided by adjacent particle size spacing.

For the suspensions' preparation, the selected polydisperse particles were made into a desired particle concentration and ionic strength (200 mM NaCl) with deionized water, and then, the suspension was sonicated for 2 h to achieve a thorough dispersion.

### 2.2. Porous Medium

Natural sand is a typical experimental material and was used as the porous medium for filling the column. The natural sand was sieved for uniformity. The PSD results showed that the porous medium material had a mean size of 0.503 mm with a standard deviation of 0.105 mm, which indicated that the sandy material was homogeneous. The bulk density of the sand medium was measured as 1.61 $g/cm^3$, and the specific density was 2.56 $g/cm^3$. The porosity of the clean porous medium was measured as 0.378 ($\pm 0.003$).

Prior to use, the sand was thoroughly cleaned by means of tap water, deionized water, HCl, and NaOH solution, in order to remove impurities. The zeta potential of the sand surface was measured to be 18.6 mV in 200 mM NaCl.

### 2.3. Experiments for the Transport and Deposition of Polydisperse Particles

A Plexiglas sand column, with a length of 50 cm and an inner diameter of 5 cm, was wet packed with the sand medium (Figure 2). Therefore, the value of each pore volume (PV) was 370.912 mL at a constant Darcy velocity of $3.51 \times 10^{-5}$ m/s, and this velocity is the average value typically used in most experiments [23]. Sixteen piezometers were unevenly installed along the length of the sand column to monitor the pressure variations during the injection processes. The screen was placed at the bottom to prevent the movement of the medium, and a dampener was installed at the top of the column to protect the sandy material from the energy of water applied. The top of the sand column was connected with a peristaltic pump to represent a suspension recharge, and a motorized stirrer operated to keep the suspension stable.

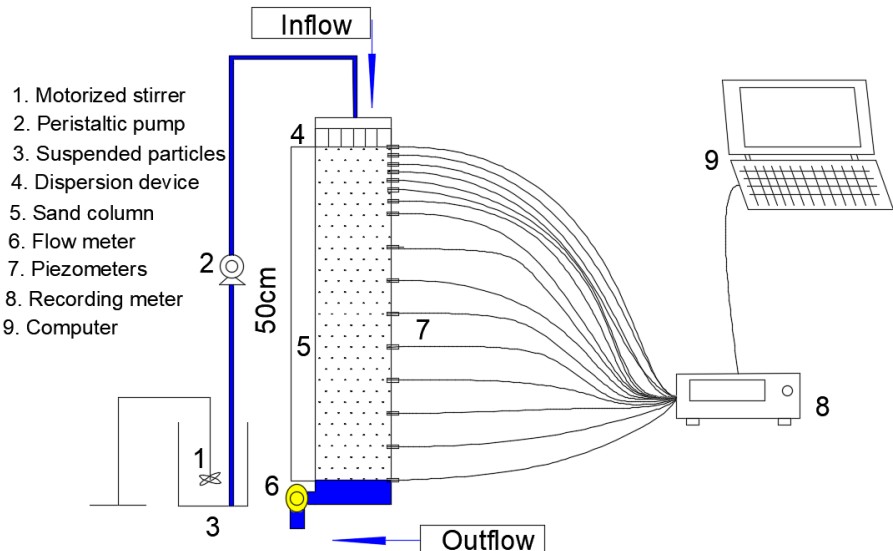

**Figure 2.** Illustration of the column experiment.

Before the particle injection, the sand column was first fed with 8 PVs of DI water and followed by 4 PVs of electrolyte solution (200 mM NaCl). Then, a total volume of 28.8 pore volumes of polydisperse particles, with a particle concentration of 300 mg/L, was injected into the sand column with five replicates during the experiments. The injected particle concentration was targeted at 300 mg/L due to the typical stormwater concentration from a review of international data [31]. The effluent concentration of the sand column was monitored with a spectrophotometer, and the hydrodynamic dispersions were determined through fitting the breakthrough curves.

During the whole period of the experiments, pressure variations due to particle capture along the column were monitored to avoid piezometer heads below the position head, and the case of an unsaturated zone was not considered.

At the end of the experiment, the mixed deposition and sandy material packed in the column was carefully excavated and cut into sections to obtain the final deposition profile of the retained particles along the column length according to the procedure described in [32]. PSD analyses were conducted for the retained particles.

### 2.4. Modeling for the Transport and Deposition Processes of Polydisperse Particles

The transport and capture processes of polydisperse particles, in a one-dimensional sand column, can be described by the advective diffusion equation with a deposition term, as follows [33,34]:

$$\frac{\partial C}{\partial t} = D \frac{\partial^2 C}{\partial z^2} - u \frac{\partial C}{\partial z} - \frac{\rho_p}{\varepsilon} \frac{\partial S}{\partial t} \tag{1}$$

$$\frac{\rho_p}{\varepsilon} \frac{\partial S}{\partial t} = k_0 \varphi C \tag{2}$$

where $C$ (M·L$^{-3}$) is the aqueous phase concentration of the polydisperse particles, $D$ (L$^2$·T$^{-1}$) is the hydrodynamic dispersion coefficient, $z$ (L) is the space coordinate, $S$ (L$^3$·L$^{-3}$) is the solid phase concentration of retained particles, $u$ (L·T$^{-1}$) is the Darcy velocity, $t$ (T) is the time coordinate, $\varepsilon$ is the porosity of the sand porous medium, and $\rho_p$ (M·L$^{-3}$) is the bulk density of the particles.

The kinetic equation of the polydisperse particles is given as follows [32]:

$$k_0 = -\frac{3(1-\varepsilon)}{2d_c} u \int_0^\infty Q(x) ln(1 - \eta(x)) dx \tag{3}$$

where $k_0$ (T$^{-1}$) is the initial deposition coefficient, $d_c$ (L) is the collector diameter, and $Q(x)$ (L$^{-1}$) is the mass density of the polydisperse particles. $\eta(x)$ is the capture probability (collector efficiency) of a particle with a diameter of $x$. $x$ (L) is a collection of particle sizes. Details about the seven existing models to calculate the capture probability can be found in the Supplementary Materials. If the particles are assumed to be monodispersed, and $x$ is a constant value, $\int_0^\infty \eta(x) dx = 1$, and Equation (3) is expressed as follows:

$$k_0 = -\frac{3(1-\varepsilon)}{2d_c} u ln(1 - \eta) \tag{4}$$

Equation (4) is the expression of a constant first-order deposition coefficient, which is derived from the classical CFT [35]. Thus, the deposition coefficient of the monodispersed particles is a special case of the polydisperse particles.

$\varphi$ is a dimensionless particle deposition function, and this dimensionless model is given as follows [36]:

$$\varphi = \left(1 - \frac{S}{S_m}\right) \tag{5}$$

where $S_m$ is the maximal retention ($S_m \leq \varepsilon$), and measured to be 0.265 according to the methods described in [32].

The mass density of particles will constantly change, as particles of different capture probabilities are arrested by the porous medium. The mass density is given as follows:

$$Q_{out}(x) = \frac{(1 - \eta(x))^{\varphi \frac{3(1-\varepsilon)}{2d_c} \Delta l} Q_{in}(x) C_{in}}{C_{out}} \tag{6}$$

where $Q_{out}(x)$ (L$^{-1}$) is the mass density of the polydisperse particles and $C_{in}$ and $C_{out}$ (L$^3$·L$^{-3}$) are the inlet concentration and outlet concentration. $\Delta l$ (L) is the length of a finite element of the sand column.

The numerical solutions of $C_{out}$ and $S$ for the coupled partial and ordinary differential Equations (1)–(6) can be achieved by an explicit algorithm, which requires a set of appropriate boundary and initial conditions.

A constant concentration of the polydisperse particles is assumed at the inlet of the sand column, and the corresponding boundary condition is as follows:

$$C = C_0 \ at \ z = 0, t > 0 \tag{7}$$

A second boundary condition with no change of concentration at the sand column exit was adopted.

$$\frac{\partial C}{\partial z} = 0 \ at \ z = L, t > 0 \tag{8}$$

where $C_0$ (M·L$^{-3}$) is the initial concentration and $L$ (L) is the column length.

The initial conditions corresponding to a clean sand porous medium are as follows:

$$\begin{aligned} C(z,0) = 0 \\ S(z,0) = 0 \end{aligned} \tag{9}$$

The numerical solution of $C$ and $S$ can be achieved by setting $\Delta t$ = 2 min and $\Delta l$ = 0.5 cm. There were 1262 time steps needed to complete the simulations, and an estimation of the discretization error (compared with $\Delta t$ = 1 min and $\Delta l$ = 0.1 cm) was about 0.52%. Therefore, the options of $\Delta t$ = 2 min and $\Delta l$ = 0.5 cm were acceptable when a trade-off was made between the time cost and efficiency.

*2.5. Evaluation Criterion*

In order to compare and analyze the differences between the model results and the measured results, the standard mean relative error was used as the calculation method of the error in the data analysis, and the calculation formula is as follows:

$$SMRE = \frac{\sum_{i=1}^n |y_i - y_i'|}{n(y_{max} - y_{min})} \tag{10}$$

$$RMSE = \sqrt{\frac{1}{n} \sum_{i=1}^n (y_i - y_i')^2} \tag{11}$$

$$MAE = \frac{1}{n} \sum_{i=1}^n |y_i - y_i'| \tag{12}$$

where $SMRE$ is the standard mean relative error, $RMSE$ is the standard mean relative error, and $MAE$ is the mean absolute error. $y_i, y_i'$ are the measured data and simulated data, respectively; $y_{max}, y_{min}$ are the maximum value and the minimum value of the measured data; $n$ is the number of data.

**3. Results**

*3.1. XDLVO Energy*

The zeta potentials of the polydisperse particles and the sand surface were both negatively charged in deionized water, but a 200 mM NaCl solution was added to enhance attachment in filtration by adsorption to produce charge neutralization and (or) bridging. The interaction between the polydisperse particles and the sand surface were calculated to be attractive in the high ionic strength solution, which indicated a favorable condition. The XDLVO theory was employed to calculate the interaction energy between the polydisperse particles and the sand surface, and plate–plate interactions were assumed. The XDLVO forces include van der Waals attraction, electric double-layer repulsion, and Lewis acid–base interactions [37]. The XDLVO energy profile between the polydisperse particles and sand medium is presented in Figure 3. A primary minimum was observed to exist under

the experimental conditions. It is easy for the particles to attach to the surfaces of the sand medium in a primary minimum and almost unlikely to be released into the suspension [38].

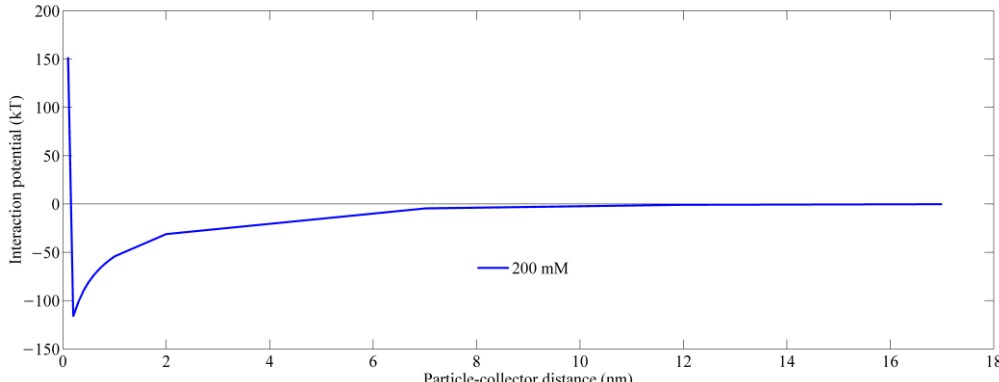

**Figure 3.** XDLVO energy profiles for the polydisperse particles and sand medium in 200 mM NaCl solution.

### 3.2. Seven Correlation Equations for Polydisperse Particles

The results of seven correlation equations for polydisperse particles are summarized in Figure 4. The capture probability of each particle is plotted as a function of the particle diameter. It was also observed that polydisperse particles of different sizes were captured by the porous medium according to their capture probabilities. The general trend of the results calculated by all seven correlation equations were similar, and there existed a particle size with a minimum opportunity for removal. Nevertheless, this critical particle size varied with different models. For the NG and MHJ models, this critical particle size was about 4 μm, while this critical particle size was about 1 μm for the other five models. For particles larger than the critical value, the capture opportunity increased rapidly with the particle size, and removal was mainly accomplished by sedimentation and (or) interception. For particles smaller than the critical value, the capture opportunity increased with decreasing particle size, and removal was mainly accomplished by diffusion.

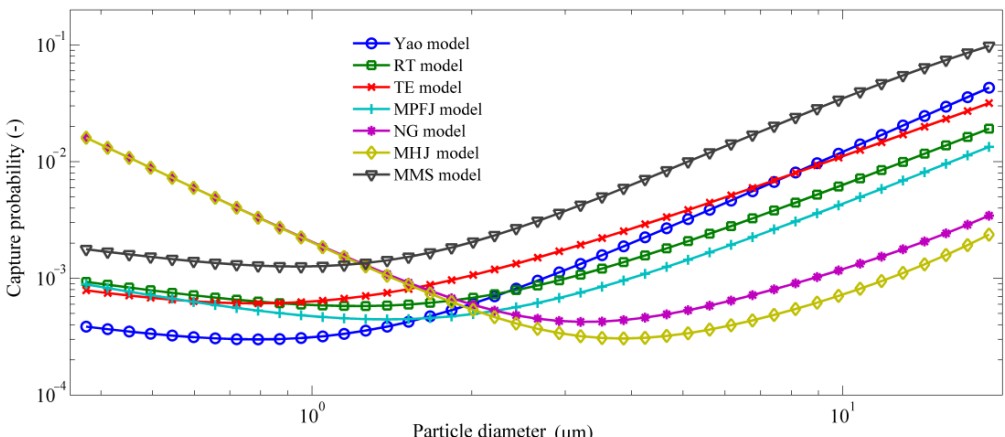

**Figure 4.** Comparison of the calculated capture probability for the polydisperse particles (0.375 to 18.863 μm) with the seven existing models. Data: $\varepsilon = 0.378$, $u = 3.51 \times 10^{-5}$ m/s, $\rho_p = 2.53 \times 10^3$ kg/m$^3$, $d_p = 2.932 \times 10^{-6}$ m, $d_c = 5.03 \times 10^{-4}$ m, $D = 2.31 \times 10^{-6}$ m$^2$/s.

The Yao model is just the superposition of three capture mechanisms due to Brownian motion, gravity, and interception, assuming the hydrodynamic effect is balanced by the van der Waals force. The RT model, TE model, MPFJ model, NG model, and MHJ model all take into account the joint effects of the van der Waals force and the hydrodynamic retardation in addition to the three mechanisms. The RT model mainly focuses on non-Brownian

particles (>1 μm), and the MHJ model is suitable for particles in low fluid velocity. The six models mentioned above neglect the mutual interactions of different capture mechanisms, while the MMS model considers the interplay between the different capture mechanisms.

Furthermore, the NG and MHJ models calculate capture opportunity, which was about an order of magnitude larger than the other models when the particle size was less than 1 μm. On the contrary, the two models predicted a much smaller capture opportunity when the particle size was larger than this critical particle size. Overall, the MMS model predicted a larger capture probability over the particle size range, which could produce a higher deposition coefficient.

### 3.3. The Transport of Polydisperse Particles in the Porous Medium

The breakthrough curves (BTCs) of the column experiment data simulated by the seven models are represented by the relative concentration as a function of the injected NVp in Figure 5. The BTCs are the relative concentration curve (ratio of the outlet concentration $C$ to the inlet concentration $C_0$) over time. The simulated BTCs of all the models showed a similar behavior. The effluent concentrations increased with time (injected pore volume) for the whole period of the experiments. After the 28.8 Vp of polydisperse particles injection, finally, the relative concentration reached a steady-state plateau value of 0.833. These models predicted a higher relative concentration of the outlet at the early stage of injection (NVp < 5) except for the MMS model, and the MMS model predicted the steady-state concentration plateau at 10 Vp, which was earlier than the other models.

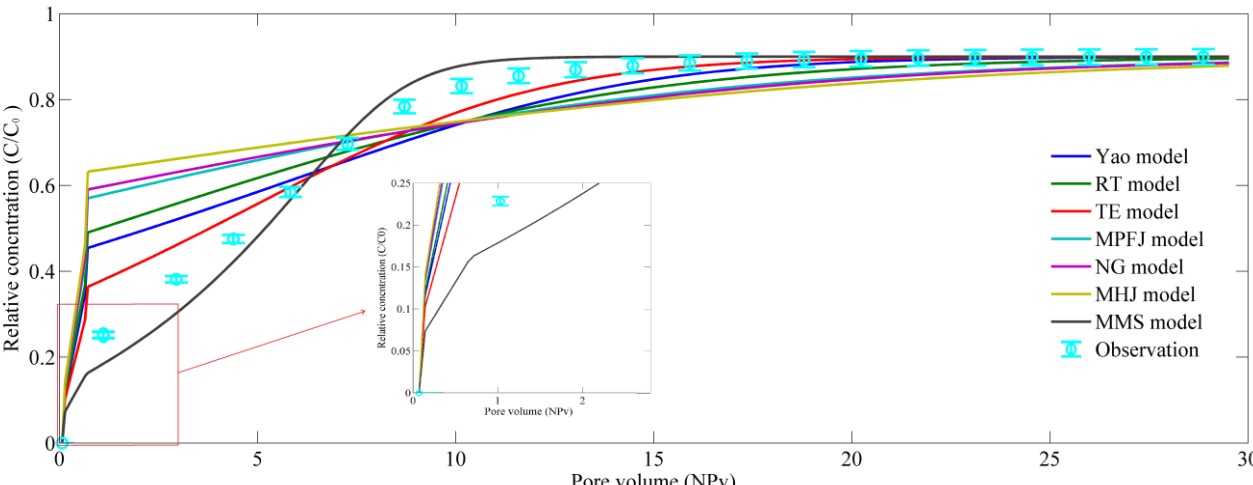

**Figure 5.** Comparison between the observed breakthrough curve (symbols) and simulated BTCs (solid line) with the existing models for the polydisperse particles, plotted as a function of the number of pore volumes (NPv). The mean concentration of five measurements, with the 95% confidence interval, is presented.

Figure 6 shows the normalized Taylor diagram of the relationship between the simulated values of the seven models and the experimental results. The closer to the observation point, the better the simulation of the model. The seven models all seemed to portray the transport and capture processes of the polydisperse particles in terms of the BTCs well, since the correlation coefficients of the seven models were more than 0.9 and the root-mean-squared error was between 0.1 and 0.5, which are generally acceptable results for BTC predictions. The MMS model was the best under the experimental conditions, and its correlation coefficient and root-mean-squared error were 0.99 and 0.1, respectively.

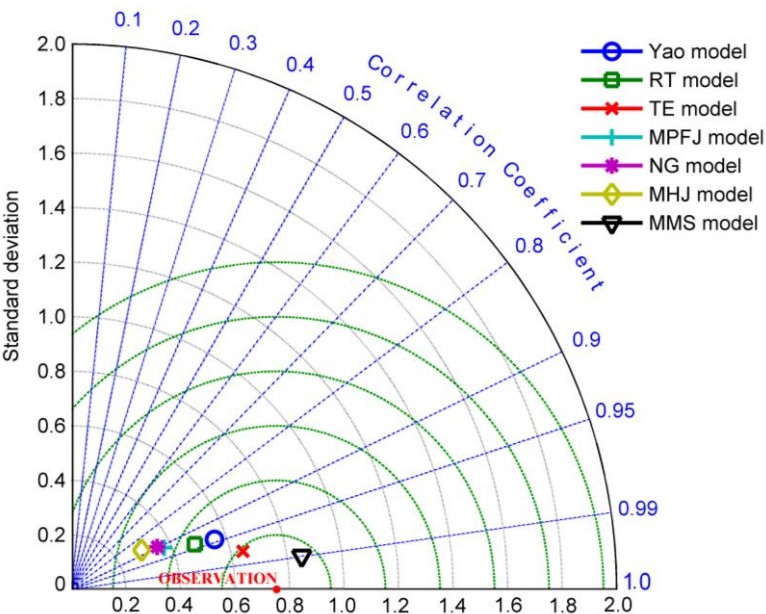

**Figure 6.** Taylor diagram of simulated values with seven existing models and observed values.

### 3.4. The Final Retention Profile

The final retention profile is also a pivotal indicator to evaluate a model's ability to characterize the fate of polydisperse particles in the saturated porous medium. Figure 7 exhibits the deposition profiles of the experimentally observed data and simulated results along the sand column at the end of the experiments. The retention profile of the experimental data was a hyper-exponential profile characterized by two distinct sections, with a steeper slope at the upper segment of the porous medium, followed by a relatively flatter slope at the lower part of the column. The experimentally observed hyper-exponential profile indicated a higher proportion of the polydisperse particles were captured near the entrance of the sand column, and much fewer particles were captured near the bottom of the column. Two models stood out, namely the Yao model and the MMS model, as both were able to depict the hyper-exponential profile of the retained polydisperse particles under the experimental conditions (Table 2). However, the rest of the models tended to homogenize the final retention profile of the captured particles and failed at the attempt to capture this obvious feature.

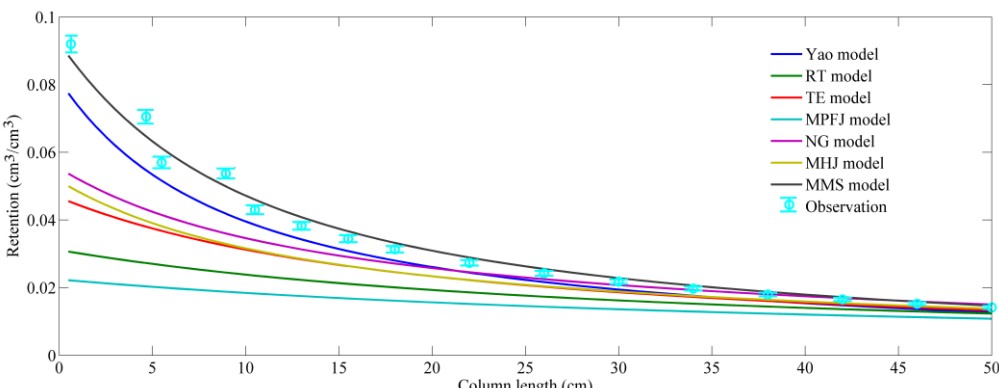

**Figure 7.** The comparison of the experimentally observed and simulated retention profiles for polydisperse particles along the sand column.

**Table 2.** The standard average relative error of each model for the particle retention profiles.

| Model | Yao Model | RT Model | TE Model | MPFJ Model | NG Model | MHJ Model | MMS Model |
|-------|-----------|----------|----------|------------|----------|-----------|-----------|
| SMRE | 0.009 | 0.044 | 0.024 | 0.058 | 0.013 | 0.017 | 0.005 |
| RMSE | 0.003 | 0.015 | 0.010 | 0.019 | 0.007 | 0.009 | 0.002 |
| MAE | 0.058 | 0.123 | 0.099 | 0.138 | 0.083 | 0.095 | 0.047 |

It was shown that the discrepancies between the existing seven models for polydisperse particles presented significant effects on the laboratory-scale applications of the CFT under favorable conditions. Both the BTC results and retention profiles demonstrated the ability of the MMS model to characterize the processes of polydisperse particles' transport and deposition under the experimental conditions. At a higher injection concentration (300 mg/L), the interaction between different deposition mechanisms might not be ignored [39]. The MMS model stood out since it can not only model interception, sedimentation and diffusion, but also the mutual interactions among these three mechanisms. Thus, the results of the MMS model were employed to analyze the distribution of the particle deposition coefficients, both spatially and temporally.

## 4. Discussion

### 4.1. The Formation of a Hyper-Exponential Deposition Profile

Under favorable conditions (particle–surface lacking net repulsion), the final particle retention profiles were observed to be exponential in saturated porous media, and it is generally assumed that, once the particles are immobilized by the porous surface, they will never be released back in to the bulk solution [40,41]. However, a hyper-exponential deposition profile was observed to exist under favorable conditions, which was inconsistent with the classical CFT predictions. Figure 8 presents the spatial and temporal distribution of the accumulated retention during the whole stage of the experiments simulated by the MMS model. The results of the MMS model showed an uneven distribution of particle retention both temporally and spatially. This uneven distribution increased with increasing injected polydisperse particles, and finally, the hyper-exponential deposition profile formed along the sand column.

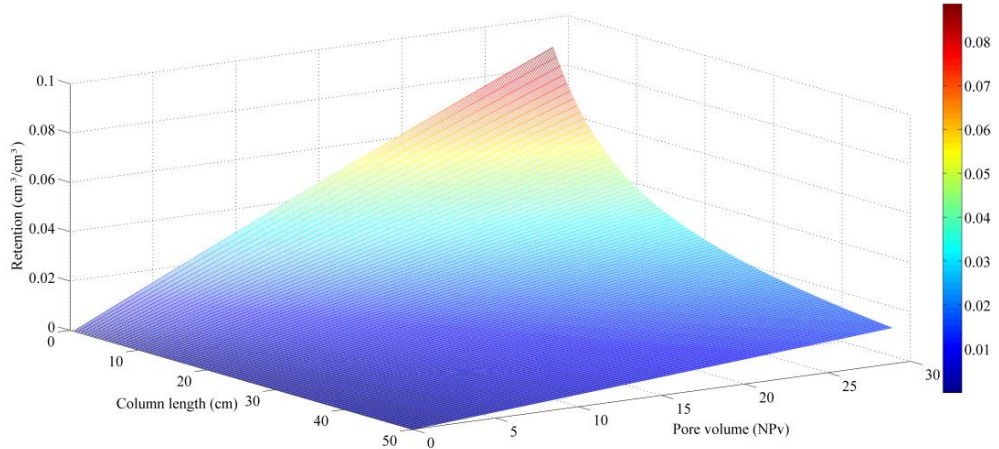

**Figure 8.** Spatial and temporal distribution of particle retention simulated based on the MMS model.

The classical CFT-predicted retention profile is log-linear with the column depth under favorable conditions, if the polydispersity of the suspended particles is ignored [42]. The hyper-exponential profile is usually attributed to unfavorable conditions, where the like-charged particle and medium surface generate a repulsive barrier that stops the particle from approaching the collector surface [9]. Under favorable conditions, the formation of hyper-exponential profile should be attributed to the particles' polydispersity. It is obvious that particles of different sizes are unevenly arrested by the porous medium according

to their capture probabilities (Figure 4). The capture probability for particles larger than the critical particle size increased sharply with the particle diameter, but the increase was much slower for particles smaller than the critical size (the MMS model in Figure 4). This distinction implies that the particle capture probability will disproportionally increase with the particle size. With higher capture probabilities, more of the larger particles are captured near the entrance of the porous medium, while more of the smaller particles with lower capture probabilities tend to be transported through the porous medium.

### 4.2. Spatial and Temporal Evolution of Deposition Coefficient

The uneven distribution of particle retention suggested decreased distributions of the deposition rates. The hyper-exponential retention profiles could be directly explained by the spatial and temporal distribution of the deposition coefficients, which is shown in Figure 9. The results also showed that, as uneven particle capture occurred, the distribution of the particle deposition coefficients varied substantially in time and space. The uneven distributions of the deposition coefficient were consistent with the spatial distribution of particle retention (Figure 8). The deposition coefficient declined with the column length even at the beginning of the injection experiment, rather than a constant first-order deposition rate, which is usually the assumption for CFT. The kinetics of polydisperse particle deposition on the porous surface decreased with length, as particles with bigger capture probabilities were retained by the upper part of the sand column and the rest of the particles with smaller capture probabilities were too difficult to be arrested by the lower part of the porous medium. In this case, the porous medium at the lower part of the sand column suffered a small drop of the deposition coefficients, and the deposition coefficients looked unchanged, as simulated by the MMS model in Figure 9.

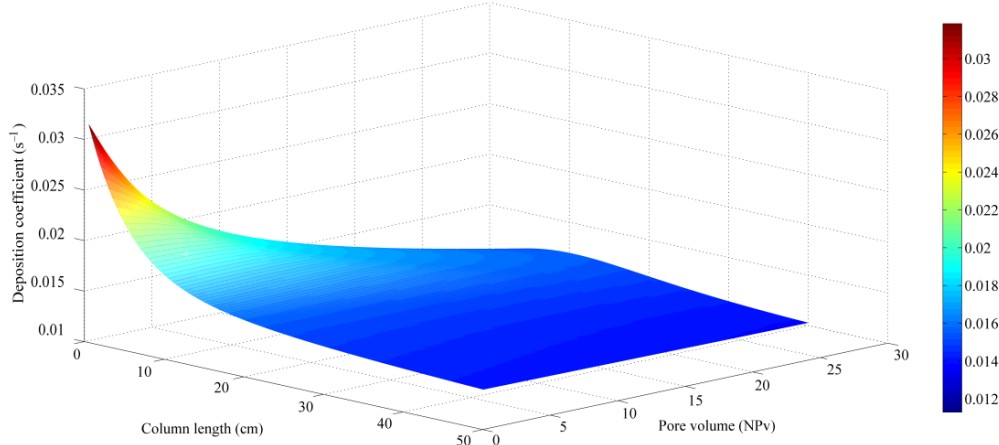

**Figure 9.** Spatial and temporal distribution of the deposition coefficients, simulated based on the MMS model.

### 4.3. Particle Size Distribution Analysis of the Retained Particles

Figure 10 presents the PSDs of the retained polydisperse particles along the column length at the end of the experiment. The PSDs varied from the inlet of the sand column to the outlet. The upper part of the porous medium ($z \leq 10$ cm) possessed almost the same particle size range as that in the injected inlet particles (0.375 to 18.863 μm). The PSD of the retained particles gradually narrowed, as larger particles with higher capture opportunities were retained upstream of the smaller particles, which were more likely to be transported further. The particle size range of the sand bottom ($40 \leq z \leq 50$ cm) was less than 8 μm. The measured median particle size also declined from the top to the bottom. The maximum ratio of the particle size to the mean collector diameter ($d_p/d_c$) was 0.037, much smaller than the critical value of 0.05. This ratio implies that mechanical straining is not a primary factor responsible for the particle retention [43,44]. The deposition of large particles is still subject to interception and sedimentation due to gravity and their sizes, while Brownian diffusion

governs the fate of smaller particles [19]. Further, the mutual interactions among these three mechanisms can also contribute to the deposition of the polydisperse particles [25].

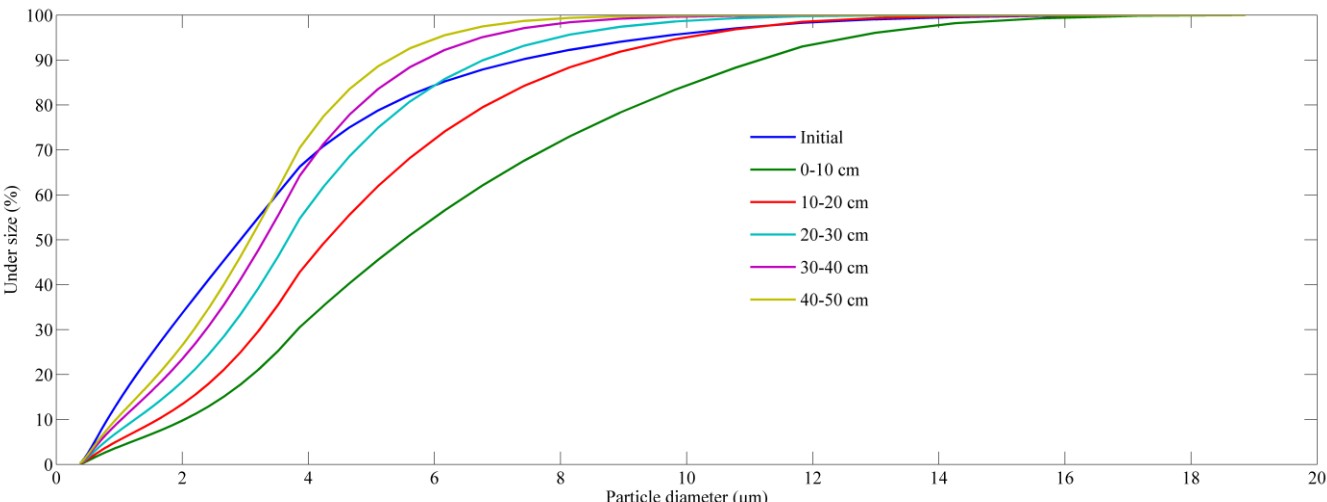

**Figure 10.** Particle size distribution (PSD) analysis of the retained polydisperse particles along the sand medium.

*4.4. Spatial Distribution of the Mass Density*

When the polydispersity of suspended particles in groundwater cannot be neglected, the polydisperse particles result in a distinct deposition behavior [45]. The spatial distribution of the particle mass density along the column is manifested in Figure 11. Particles of different sizes have different capture probabilities; thus, the removal of polydisperse particles was uneven along the column length. Because the dynamic light scattering method can only measure the relative proportion of the outlet-recovered particles, if the same proportion of particles as in the influent is removed, the mass density curve will stay the same. The mass density of a certain particle decreases if it has a greater probability to be captured than other particles. On the contrary, the mass density of a certain particle increases if it is removed by less in proportion. According to the evolution of the mass density, three types of particles were identified: (1) small particles, the size range being (0.375, 1.941 μm) and the mass density increasing along the length (Figure 12a); (2) middle particles, the size range being (2.123, 8.715 μm) and the mass density increasing, then decreasing (Figure 12b); (3) large particles, the size range being (9.732, 18.863 μm) and the mass density continuing to decrease (Figure 12c). The mass densities of three typical particles (d = 0.545, 3.206, 10.779 μm) are plotted in Figure 12. It can be seen that the mass density curve of the large particles dropped sharply, while the curves of the small and middle particles rose at first, because a bigger proportion of the large particles were removed and the relative proportion of the small particles and middle particles increased. However, as the particles reached the deeper porous medium, almost all large particles were removed and more middle particles were removed, and the mass density curve of the middle particles started to decline, while the curve of small particles kept moving upward. The variations of the particle mass densities reflected the non-uniform retention of the polydisperse particles. The evolution of the mass density was consistent with the PSDs of the retained polydisperse particles along the column length (Figure 10).

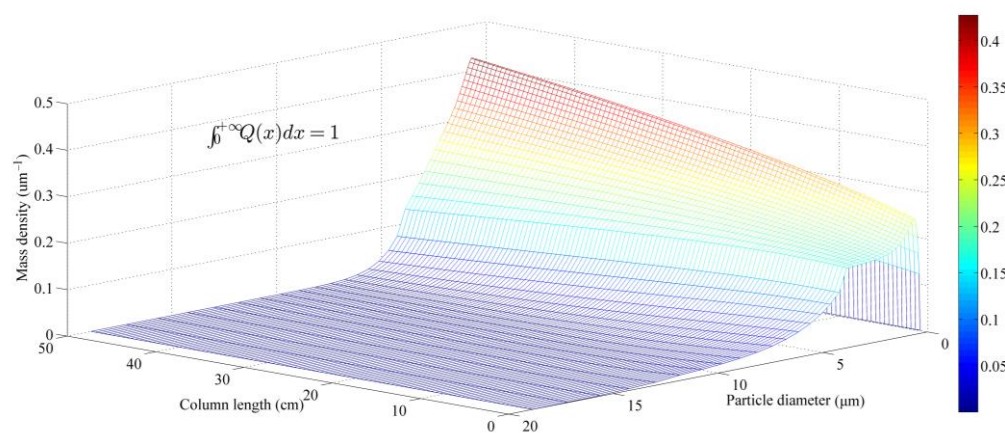

**Figure 11.** Spatial distribution of the mass density for the aqueous-phase polydisperse particles along the length of the sand column.

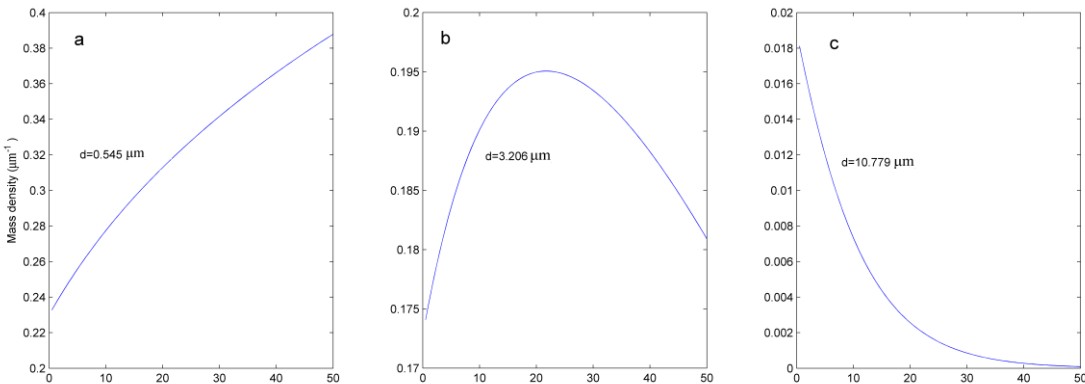

**Figure 12.** Evolution of the mass density for three typical particles along the length of the column: (**a**) $d_p$ = 0.545 μm, (**b**) $d_p$ = 3.206 μm, (**c**) $d_p$ = 10.779 μm.

## 5. Conclusions

The CFT predominately focuses on bacteria, viruses, and colloids that are about 1 μm in size or smaller. These suspended particles are not only small, but also monodispersed, and Brownian diffusion is the operative transport mechanism. However, a wide size range of suspended particles is ubiquitous in natural groundwater environments, and the impacts of particle polydispersity on particle deposition processes are often neglected. In this study, the mass density (the probability density of the particle size distribution) was incorporated into the depth-dependent deposition rate to account for the implications of particle polydispersity. Particles with different diameters have different capture probabilities depending on the particle size. Large particles generally with high capture probabilities tend to be captured by the porous medium, while small particles with low capture probabilities tend to be transported throughout the medium. This uneven capture of the polydisperse particles caused the formation of a hyper-exponential deposition profile, even under favorable conditions. At a higher injection concentration (300 mg/L), the particles were densely populated, and all the deposition mechanisms might operate jointly. The MMS model takes into account the mutual interactions among these mechanisms, which makes it more suitable to portray the transport and deposition processes of the polydisperse particles in the saturated medium.

It is also important to note that our sand column experiments of the polydisperse particles injection is just an exploration. Thus, a comprehensive testing of the existing models will require new experiments at physical and chemical conditions affecting particle migration in porous media, especially a pore-scale investigation of the impacts of particle polydispersity on particle transport and capture processes in a saturated medium.

**Supplementary Materials:** The following Supporting Information can be downloaded at: https://www.mdpi.com/article/10.3390/w15122193/s1, Table S1: Summary of dimensionless parameters present in the existing models; Table S2: List of the existing equations for comparison. See Table S1 for the parameters' definition.

**Author Contributions:** Conceptualization, L.F. and L.Y.; methodology, W.L. (Wenbing Luo); validation, Z.Z. and L.Y.; data curation, Y.L.; writing—original draft preparation, Z.Z.; writing—review and editing, S.N.; supervision, W.L. (Wei Li); project administration, Y.L.; funding acquisition, Z.Z. and Y.L. All authors have read and agreed to the published version of the manuscript.

**Funding:** This study is funded by the National Natural Science Foundation of China project (No. 42202276). The research work is also financially supported by the Knowledge Innovation Program of Wuhan-Shuguang (No. CKSD2022363/NY) and the fund for Basic Scientific Research Business of Central Public Research Institutes (No. CKSF2021452/NY).

**Data Availability Statement:** Data will be made available upon request.

**Acknowledgments:** The authors acknowledge valuable comments from the Reviewers, which led to a significant improvement of the paper.

**Conflicts of Interest:** The authors declare no conflict of interest.

## Nomenclature

| | |
|---|---|
| $C$ | aqueous phase concentration of the polydisperse particles |
| $C_{in}$, $C_{out}$ | inlet concentration and outlet concentration |
| $D$ | hydrodynamic dispersion coefficient |
| $z$ | space coordinate |
| $t$ | time coordinate |
| $S$ | solid phase concentration of retained particles |
| $S_m$ | maximal value of $S$ |
| $u$ | Darcy velocity |
| $\varepsilon$ | porosity of the sand porous medium |
| $\rho_p$ | bulk density of the particles |
| $k_0$ | initial deposition coefficient |
| $d_c$ | average collector diameter |
| $Q(x)$ | mass density of the polydisperse particles |
| $\eta(x)$ | capture probability of a particle with a diameter of x |
| $x$ | a collection of particle sizes |
| $\varphi$ | a dimensionless particle deposition function |

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
