# Peer review of "The Comparison of Seven Models to Simulate the Transport and Deposition of Polydisperse Particles under Favorable Conditions in a Saturated Medium"

_water, doi:10.3390/w15122193_

Round 1

Reviewer 1 Report

The authors compared the applicability of seven typical models using the results of sand column experiments conducted by themselves. This work is of interest to this journal. However, it will benefit from some revisions. Specific suggestions or comments are as follows:

1.     Tabel 1. ‘Error! Reference source not found.’ Please check it.

2.     Lines 350-352. ‘The maximum ratio of the particle size to the mean collector diameter (dp/dc) was 0.037, much smaller than the critical value of 0.005.’ Please check it.

3.     According to the evolution of the mass density, three types of particles were identified (i.e., small, middle and large particles). How about the relationship between the size of these particles and aperture of the flow channel? For example, when the ratio of particle size to channel aperture reaches what level, particles will be intercepted.

4.     The comparison of the seven models should be more comprehensive, that is, the applicable conditions and deficiencies of each model should be explained.

Overall, the author's English writing proficiency is good, but detailed checks are needed to avoid some minor errors.

Reviewer 2 Report

General comments:

The manuscript deals with a topic of great interest since the transport of monodisperse particles has been largely studied but has its limitations when applied to practical applications. The authors apply a polydisperse particle retention model based on a stochastic method to analyze different expressions for the initial collector efficiency. A comparison with experimental data is carried out. The manuscript is well structured but there are some methodological and formal aspects that should be corrected before publication. Therefore, my recommendation is to accept the manuscript after major revisions have been carried out.

Major points:

1.     There is an inconsistency in the methodology employed. Eq. (1) is the classical advection-diffusion equation with a deposition term. But the use of Eq. (3) expressed in terms of Q(x)ln(1-eta)) was originally derived from the single steady non-diffusive equation (Eq. (1) in your reference [30]). So, why should it be a valid expression for your Eq. (1)?

2.     The original work [30] from which the main formulation shown in the paper is based on has some dimensional inconsistencies. This can be clearly seen in your Eq. (6) in which the exponent of the (1-eta) term has units of length when, obviously, it should be dimensionless. The same happens in the original work (Eq. (5) and (7) in your reference [30], in which the exponent is not dimensionless). Please, revise the method to make it dimensionally consistent and physically correct in all the steps.

3.     Line 189: What is Qout(d)? Which is the difference between (d) and (x) in Eq. (6)? On the other hand, Eqs. (1)-(2) are global equations in terms of particles but you work with the local term Q(x). How is Q(x) related to C and to S?

4.     Line 191 -192. You should include a detail description of the numerical method (commercial vs in-house, method, discretization error (space and time), etc.)

Minor points:

1. Line 44: Substitute “et al.” by the list of the other mechanisms.

2. Line 45 and throughout the text (including figures): units written “um” should be written using the Greek letter mu.

3. Line 87. Table 1. There are links to references not found. Also, the “Transport mechanisms” column could be simplified by using some abbreviations of the terms used (that could listed at the foot table). This change would make the table more readable.

4. Line 89: “Couple of mysteries remain veiled”. Please, specify them.

5. Line 128: It would be interesting to add more details about the sand material: D10, D60 and coefficient of uniformity.

6. Line 139: Darcy velocity? Do you mean superficial velocity? Why has this velocity value been chosen? Under which criterion? Besides, use the symbol X before the power term (also in other terms as in the expression of diameters, density, etc.).

7. Line 139 and 155-156: Piezometers: pressure data have not been used in the manuscript. Why not?  Besides, it would be interesting to know more technical information of the piezometers and the pressure acquisition system.

8. Line 158-161: Please, specify the size of the disc and the filtration characteristics of the filter used to retain the sand when washed.

9. Line 168: Reference [33] is for monodisperse particles. Add references that really support the validity of Eqs. (1)-(2) for polydisperse particles.

10. Line 178 and throughout the text: Please use the correct superscripts in the dimensions of variables (ex. L-1 => L^-1)

11. Line 180: The supplementary material proposes different expressions for Eta_0. However, here you sue Eta(x). Please unify the nomenclature.

12. Line 182: The text: “… and Equation (2) is expressed …” is wrong since you refer to Equation (3) instead.

13. Line 187: Which is the value of the maximal retention Sm used? And which has been the criterion to select it?

14. Line 197: You use exactly the same evaluation criteria as in reference [30]. However, other comparison methods should be made. I recommend testing other statistics such as MARE (mean absolute relative error) or RMSrE (root mean square relative errors).

15. Line 244: Define the concept of breakthrough curve.

16. Line 339. Fig. 9: The deposition coefficient is almost unchanged with time. Please, discuss why should it be like this.

17. Supplementary material: A nomenclature section should be included since many variables are not described. In the MMS model, the exponents (1) and (2) are real exponents of simply numbers for the footnotes? If the later, please change them by (a) and (b) to avoid confusion.

Round 2

Reviewer 2 Report

I strongly encourage the authors to carefully review their answers before sending to the reviewer. There are many obvious mathematical errors that I have simply decided not to continue with my revision. 

For example: in Response 1, Eq. (4), a minus sign is missing. In response 2, Eqs. (7) and (8), Nc has units of m^-3 but then in Eq. (9) it is dimensionless. Eq. (4) in the new manuscript has an expression different from the Eq. (4) in the original document, which is very different from the classical expressions and nothing is said. 

So my recommendation is to thoroughly revise your answers (no mathematical flaws) to my previous queries and submit them again. 

Round 3

Reviewer 2 Report

The authors have not detected the mathematical error they made in their first answer to my point #2. The expression Nc in equations (7) and (8) in page 5-6 of your “Response 2” has units of L^-3 and it is not dimensionless. Note that in the supporting information of Johnson and Hilpert that you have provided, this term is written as Nc* (units L^-3 in equations SI-2 and SI-3) and it is different than Nc (dimensionless in equations SI-4 and SI6). In your answer, you used the same symbol Nc for all equations, which was wrong. This error does not invalidate your final equation but you should be very cautious when carrying out this kind of mathematical developments.

Besides that, the authors have satisfactorily answered most of my previous queries. As a consequence, my recommendation is to accept the manuscript after a minor revision has been carried out. I think that this minor revision will help the readers to better understand authors’ work and to enhance its dissemination.

Minor points:

1) Please, include a nomenclature section in the main manuscript. It will help the readers to follow your work.

2)   In my first review, when I proposed to correctly apply superscripts, I did not mean to explicitly type the ^ symbol. Since I wrote my report in plain text, the ^ symbol in my example meant “superscript”. So, please, delete the ^ symbol in all terms.

3)      In line 239 (numerical simulation), say the number of time steps needed to complete the simulations and comment the % error in some of the reported values when changing the Dt to 1 min and Dl to 1 cm (or similar). It is important to give an estimation of the discretization (in time and space) error.
